# Mechanisms of Resistance to Novel Immunotherapies in B-Cell Lymphomas: Focus on CAR T and Bispecific Antibodies

**DOI:** 10.3390/cancers17213453

**Published:** 2025-10-28

**Authors:** Gloria Arena, Roberto Chiarle

**Affiliations:** 1Department of Molecular Biotechnology and Health Sciences, Molecular Biotechnology Center, University of Torino, 10126 Torino, Italy; gloria.arena@unito.it; 2Division of Hematopathology, IEO European Institute of Oncology IRCCS, 20141 Milan, Italy; 3Department of Pathology, Children’s Hospital and Harvard Medical School, Boston, MA 02115, USA

**Keywords:** B-cell lymphomas, bispecific antibodies, Chimeric Antigen Receptor T cell therapy, CD19-CAR T, immunotherapy, resistance

## Abstract

**Simple Summary:**

B-cell lymphomas are blood malignancies derived from B lymphocytes and associated with decreased survival rates if recurrent or refractory to treatment. Novel immunotherapies approaches, including CD19-CAR T cell therapy and bispecific antibodies, are improving patients’ survival after failure of standard-of-care treatments. Despite these promising results, the onset of molecular resistance to these cutting-edge therapeutics limits their efficacy. This review aims to describe the molecular mechanisms of resistance to CAR T cell therapy and bispecific antibodies in B-cell lymphomas, with a focus on CAR T cell fitness, T cell exhaustion, tumor-intrinsic determinants, and the role of the immunosuppressive tumor microenvironment.

**Abstract:**

Treatment paradigms for B-cell lymphomas have evolved significantly in the last decades. Nevertheless, the widespread clinical use of immunotherapy has demonstrated that it invariably leads to the development of resistance. This review outlines the underlying molecular mechanisms of resistance associated with emerging immunotherapeutic strategies, including Chimeric Antigen Receptor (CAR) T cell therapy and bispecific antibodies (BsAbs). In high-grade B-cell lymphomas, nearly 50% of patients progress following CAR T treatment due to host-related factors affecting CAR T cell proliferation and persistence, as well as tumor-intrinsic factors, such as loss of CD19 epitope expression, trogocytosis, and other genomic alterations (e.g., *CD19* mutations, chromothripsis, APOBEC mutational activity, and deletions of *RHOA*). Additional genomic and epigenetic events, including mutations, alternative splicing of *CD19*, and aberrant promoter methylation, further contribute to resistance. BsAbs, representing an off-the-shelf T-cell-redirecting strategy, have recently shown promising single-agent efficacy with a manageable toxicity profile, predominantly characterized by T cell overactivation syndromes. Similarly to CAR T cell therapy, BsAb resistance arises through diverse mechanisms, such as antigen loss, T cell dysfunction (exhaustion and regulatory T cell activation), tumor-intrinsic alterations (e.g., *TP53* mutations and *MYC* amplifications), and immunosuppressive influences from the tumor microenvironment. These findings underscore the complexity of immune evasion in B-cell lymphomas and highlight the ongoing need to optimize immunotherapeutic strategies and develop combination approaches to overcome resistance.

## 1. Introduction

B-cell lymphomas are a heterogeneous group of lymphoid neoplasms derived from B lymphocytes at various stages of differentiation and divided into Non-Hodgkin (NHL) and Hodgkin Lymphomas (HLs), including Diffuse Large B-Cell Lymphoma (DLBCL), Follicular Lymphoma (FL), and Mantle Cell Lymphoma (MCL) [1,2]. In high-income countries, the incidence of B-NHL (85–90% of NHL) is estimated at 20 new cases per 100,000 per year, while HL is a rare disease with an incidence of 85,000 new diagnoses per year [1,3]. The etiopathogenesis of B-cell NHL (B-NHL) is usually linked to reciprocal chromosomal translocations between one of the immunoglobulin loci and a proto-oncogene, which determine the constitutive expression of the translocated gene under the immunoglobulin loci [4]. In contrast, the etiology of HL remains unclear; however, HIV-related immunosuppression and Epstein–Barr (EBV) viral infection are frequently associated with this malignancy [2]. Common early symptoms of B-NHL and HL are represented by painless lymphadenopathies, fever, night sweats, weight loss, and fatigue; however, a wide range of clinical presentations may arise depending on lymphoma localization [1,2]. A biopsy sample, in association with clinical examination and laboratory and imaging studies, are needed for an accurate diagnosis, and, according to the tumor’s stage, various therapeutic regimens are recommended [1].

CD19-CAR T cell therapy is currently an available option for patients with relapsed or refractory (r/r) DLBCL, r/r FL, and r/r MCL [5]. Distinct clinical trials, such as JULIET (Tisagenlecleucel in r/r DLBCL) and ZUMA-1 (Axicabtagene ciloleucel in r/r DLBCL, Primary Mediastinal B-Cell Lymphoma, or transformed FL), have investigated the overall response rates (ORRs) following CD19-CAR T cell therapy, which were, respectively, 52% (41–62, 95% IC) and 82% (73–89, 95% IC) [6,7,8]. Despite these promising results, the trials revealed concerning percentages of relapses after CD19-CAR T cell treatment, with 21% to 35% in JULIET and approximately 50% in ZUMA-1 [8]. Furthermore, the French registry DESCAR-T, which included 550 patients with Large B-Cell Lymphomas (LBCLs), showed that the median overall survival (OS) of non-responders or relapsing patients following CD19-CAR T cell therapy was 5.2 months (4.1–6.6 months, 95% IC) [9,10]. Primary resistance to CD19-CAR T cell therapy is observed in roughly 50% of DLBCL, around 10% of MCL, and 5–15% of FL cases, while secondary resistance happens in around 40–50% of DLBCL, 40% of MCL, and 30% of FL [11]. The molecular mechanisms associated with CD19-CAR T cell therapy resistance are linked to CAR T cell fitness, tumor-intrinsic determinants, host-related factors, and tumor microenvironment (TME) characteristics [11]. Because resistance to CAR T cell therapy leads to unfavorable prognosis and predictive biomarkers are still under investigation, there is an urgent need to analyze the underlying molecular mechanisms and provide safe and more efficient CAR T cell products [11].

Bispecific antibodies (BsAbs) represent an additional therapeutic option for B-cell lymphomas and are currently recommended for the treatment of r/r DLBCL and r/r FL [5]. A real-world analysis conducted by Brooks et al. demonstrated that the ORR and complete response rate (CRR) of Epcoritamab and Glofitamab in the context of DLBCL were 51.7% and 25.4%, respectively [12]. In addition, a Phase II study led by Viardot et al. demonstrated that after one cycle of Blinatumomab, the ORR of patients with r/r B-NHL was 43% and that 19% of individuals achieved a complete response [13,14]. However, approximately 50% of patients fail to achieve remission following CD20 × CD3 BsAbs monotherapy [15]. The pivotal EPCORE NHL-1 study demonstrated that the estimated 24-month progression-free survival (PFS) for patients with r/r LBCL treated with Epcoritamab was 27.8%, whereas OS rates reached 44.6% [16]. In addition, a Phase I/II study that tested fixed-duration Glofitamab monotherapy for heavily pretreated patients with r/r MCL revealed that the median PFS was 18 months [17]. The molecular mechanisms underlying BsAbs failure in B-cell lymphomas are heterogenous and include T cell exhaustion and dysfunction, antigen escape, an immunosuppressive TME, and immune checkpoint dysregulation [18]. These novel therapeutics have paved the way towards a new era of cancer immunotherapy; however, the molecular mechanisms of resistance underlying the failure of CD19-CAR T cell and BsAbs therapy need to be elucidated. In this review, we aim to provide a comprehensive description of the molecular mechanisms of resistance to CD19-CAR T cell and BsAbs therapy, distinguishing between CAR and antibody-related factors, tumor-intrinsic determinants, host characteristics, and the role of the TME.

## 2. Immune Evasion from CD19-CAR T Cell Therapy

### 2.1. CD19-CAR T Cell Fitness

The Chimeric Antigen Receptor (CAR) was first described in the 1980s, while the approval of the first CAR T cell therapies, Tisagenlecleucel (Kymriah) and Axicabtagene ciloleucel (Yescarta), by both the Food and Drug Administration (FDA) and the European Medicines Agency (EMA), occurred in 2017 and 2018 [19]. From that point forward, several CD19-CAR T cell therapies have been approved for the treatment of B-cell lymphomas. CD19-CARs are composed of four domains: an extracellular module, a spacer portion (CD8α or CD28), a transmembrane domain, and distinct cytoplasmic modules [20]. The extracellular domain is composed of an FMC63 single-chain variable fragment (scFv), which binds CD19 on lymphoma cells [21,22]. The intracellular module is characterized by the signaling domain (CD3ζ) of the T Cell Receptor (TCR) chain with additional co-stimulatory portions (e.g., CD137, also called 4-1BB, and CD28), which vary depending on CAR products and influence T cell survival [21,22,23,24]. When antigen recognition by the scFv domain is efficient, CD3ζ is phosphorylated, and downstream signaling is further amplified by co-receptor activation, resulting in the induction of CD19-CAR-T cells’ cytotoxic activity [20].

Firstly, the efficacy of CAR T cell therapy relies on correct patient referral and successful manufacturing in association with a precise infusion window [25]. In addition, the quality of CD19-CAR T cells depends on the patient’s health status at the moment of referral, and the T phenotype is clearly affected by previous therapeutic regimens, especially those containing Doxorubicin, which negatively affects T cells’ count and quality [9,25,26]. In relation to this, conditioning regimens play a crucial role in the determination of CAR T cell effectiveness, as they are linked to lymphodepletion, eradication of immunosuppressive cell populations (e.g., T regulators, Myeloid-Derived Suppressor Cells), modulation of the TME, and elimination of homeostatic cytokine sinks (e.g., IL-2, IL-7, IL-15), in addition to increased expansion, functionality, and persistence of CAR T cells [27]. Turtle et al. demonstrated that the addition of Fludarabine to a Cyclophosphamide (Cy/Flu) conditioning regimen was linked to improved proliferation and persistence of CD19-CAR T cells in patients with r/r B-NHL and better clinical outcomes [27,28]. Furthermore, in a Phase 1/2 clinical trial that enrolled patients with aggressive B-NHL, Hirayama et al. analyzed the differences between low- and high-intensity Cy/Flu lymphodepletion (high-intensity regimen: Cy 60 mg/kg × 1 + Flu 25 mg/m^2^ × 3 or ×5; low-intensity one: Cy 30 mg/kg × 1 + Flu 25 mg/m^2^ × 3 or Cy 300 mg/m^2^ × 3 + Flu 30 mg/m^2^ × 3 or Cy 500 mg/m^2^ × 3 + Flu 30 mg/m^2^ × 3) among 48 patients who were treated with CD19-CAR T cell therapy [27,29]. Higher intensity of the Cy/Flu conditioning regimen was linked to a favorable cytokine profile, measured through day 0 MCP-1 and peak IL-7 concentrations, although the multivariable analysis demonstrated that PFS was associated with the specific cytokine profile rather than the intensity of lymphodepletion [27,29]. Thus, the biological effects of conditioning regimens play a major role compared to the intensity of the treatment, representing an additional element that could determine CD19-CAR T cell therapy resistance [27,29]. However, further studies are needed to clarify the impact of MCP-1’s and IL-7’s increased concentration on CD19-CAR T cell effectiveness and the complexity of biological effects mediated by various conditioning therapeutic regimens [27,29].

Secondly, durable responses have been observed in specific populations of CAR T cells characterized by higher proliferation rates after administration [9]. Fraietta et al. analyzed the genomic, phenotypic, and functional profiles of 41 patients with advanced, heavily pretreated, and high-risk Chronic Lymphocytic Leukemia (CLL) treated with CD19-CAR T cell therapy and found that complete-responding patients were characterized by memory-like T cells prior to treatment, such as CD27^+^ CD45RO^−^ CD8^+^ T cells [30]. In addition, higher functionality of CD19-CAR T cells was associated with the production of STAT3-related cytokines, while serum levels of IL-6 correlated with CAR T cells’ expansion [30]. Resistance to CD19-CAR T cells in this cohort of CLL patients was linked to the upregulation of pathways involved in effector differentiation, glycolysis, exhaustion, and apoptosis, while a subset of CD27^+^ PD-1^−^ CD8^+^ CAR T cells with high expression of IL-6 receptor was associated with increased therapeutic responses [30]. Thus, central memory and stem-cell-like memory T cells are related to improved clinical outcomes [25]. Furthermore, CD19-CAR T cell persistence is correlated with the CAR construct [25]. CD28 and 4-1BB co-stimulatory domains induce distinct responses, with CD28 being responsible for rapid proliferation, stronger activation of effector functions, and greater secretion of inflammatory cytokines, while 4-1BB is linked to longer persistence and durable tumor control [31,32]. Despite accelerated exhaustion due to the CD28 co-stimulatory domain, similar outcomes were observed in patients with LBCL treated with CAR T cells expressing either CD28 or 4-1BB. Thus, further studies on B-cell lymphomas are needed to define the associated risk of relapse and/or refractory disease related to distinct co-stimulatory domains [32]. The clinical trial carried out by Rossi and colleagues further demonstrates that T cell functionality influences CD19-CAR T cell response [33]. Through a single-cell analysis of pre-infusion CD19-CAR T cell products derived from 22 patients with DLBCL, FL, or MCL, they identified a heterogenous profile of engineered T cells, characterized by polyfunctional CD41^+^ and CD81^+^ subsets secreting IFN-γ, IL-8, IL-5, granzyme B, and/or MIP-1a [33]. Worse objective responses (partial or complete) were observed in patients with diminished CAR T cell polyfunctionality, suggesting that resistance to CD19-CAR T cell therapy in B-cell lymphomas is also linked to T cells’ heterogeneity and fitness prior to treatment [33].

Epigenetic modifications in T cells play an additional role in the onset of CD19-CAR T cell resistance. The *TET2* gene encodes a methylcytosine dioxygenase, which is involved in DNA methylation and plays a crucial role in hemopoiesis [34,35]. Firstly, in a Pilot/Phase I study investigating the efficacy of CD19-CAR T cell therapy in patients with CD19-positive leukemia or lymphoma, TET2 dysfunction led to altered T cell differentiation, with a central memory phenotype at the peak of expansion in a patient with CLL [36]. The study highlights a lentiviral-vector-mediated insertion of the CAR construct in intron 9 of the *TET2* gene, while the patient simultaneously presented with a hypomorphic mutation in the second *TET2* allele [36,37]. As a consequence, chromatin access to various regulators of T cell effector differentiation and exhaustion was simplified, resulting in a central memory phenotype of CD19-CAR T cells, which induced durable molecular remission in this patient [36,37]. In addition to TET2 disruption, the expression of DNA methyltransferase-3α (*DNMT3A*)-targeted genes seems to be correlated with favorable responses to CD19-CAR T cell therapy [38]. The *DNMT3A* gene plays a role in the epigenetic modifications involved in embryonic development, imprinting, and X-chromosome inactivation due to CpG de novo methylation, and its deletion in CAR T cells promotes T cell multipotency and decreases the exhaustion signature [38,39]. In a cohort of 33 patients with CLL, an RNA sequencing dataset of CD19-CAR T cell products revealed that the expression of DNMT3A-targeted genes was significantly higher in responder patients, correlating with 65- to 635-fold enhanced expansion of CAR T cells after infusion [38]. These findings demonstrate that the epigenetic profile of T cells prior to CD19-CAR T cell production influences the efficacy of this immunotherapy and contributes to the onset of resistance related to early exhaustion. Table 1 summarizes the principal characteristics of commercially approved CD19-CAR T cell therapies for B-cell lymphomas, with a focus on the molecular mechanisms underlying resistance.

### 2.2. Tumor-Intrinsic Determinants

#### 2.2.1. CD19 Antigen Loss, Modification, and Reduction

Compared to engineered TCRs, CARs are independent of MHC expression and co-stimulation; therefore, they are not influenced by loss of MHC-associated antigen presentation by tumor cells, which commonly represents a major mechanism of immune evasion from TCR-T cell therapy [21,50]. Human CD19 is a 95 kDa type I transmembrane glycoprotein that belongs to the immunoglobulin superfamily [22]. The *CD19* gene contains 15 exons and is located on the short arm of chromosome 16, encoding a protein with 556 amino acids and an FMC63 binding site encoded by exon 4 [8,22]. It represents a biomarker for B lymphocytes and follicular dendritic cells, playing a crucial role in antigen-independent development and immunoglobulin-induced activation of B cells [22]. CD19-CAR T cells require the expression of CD19 on the surface of lymphoma cells, and its downregulation represents one of the major obstacles to the success of this immunotherapy [21,50]. CD19 antigen density on lymphoma cells is a decisive element that influences CAR T cell therapy efficacy, with the need for more than 1000 antigens per tumor cell for CARs to be effective, while TCR-T cell therapy requires less than 100 peptides per antigen-presenting cell to be fully active [32]. The underlying reasons include decreased kinase recruitment by CARs, the formation of a weaker immune synapse, the diminished engagement of co-receptors, and the involvement of negative downstream regulators, despite the increased affinity of scFvs and the higher density of CARs compared with TCRs [32]. Previous studies have demonstrated that 30% of relapses from LBCL are characterized by an insufficient CD19 density threshold for efficient CAR T cell targeting; however, enhancing T cell fitness could lead to increased signal strength and a diminished antigen density threshold required for cytotoxic activity [32].

Antigen modulation may derive from genetic mutations, alternative splicing, lineage switching, epigenetic and post-transcriptional modifications, trogocytosis, hyperglycosilation, and antigen shedding [32]. Sotillo et al. described hemizygous deletions, de novo frameshifts, and missense mutations involving exon 2 in certain B-cell Acute Lymphoblastic Leukemia (B-ALL) relapses and additionally analyzed alternative splicing of CD19 gene [51]. The splicing factor SRSF3 was found to be responsible for exon 2 retention, and its level decreased in relapsing patients [51]; however, its role in B-cell lymphomas needs to be elucidated. Through CRISPR-Cas9 editing with a guide RNA targeting exon 2 in various cell lines derived from B-cell malignancies (e.g., Raji, Burkitt Lymphoma), they found that frameshift mutations originating from exon 2 skipping led to the formation of a large CD19 protein isoform unable to induce CD19-CAR T cell cytotoxicity [51]. In addition, Zhang and colleagues reported that a single mutation in exon 3 of the *CD19* gene (p.163 R-L) was associated with HGBCL relapse six months after CD19-CAR T cell administration in a single patient, whereas acquired p.174 L>V mutation in exon 3 one month after CAR T cell infusion was correlated with resistance to CD19-CAR T cell therapy [52]. The absence of p.163 R-L mutation prior to treatment suggests that immune pressure represents an explanation for this mutagenic phenomenon [52].

Even though extended CAR T cell persistence is usually related to a decreased overall risk of relapse, increased rates of antigen loss are registered in association [11]. However, this molecular mechanism of resistance does not explain the success of CAR T in patients with *CD19* monoallelic or subclonal loss analyzed through flow cytometry before treatment and in a patient who carried the subclonal *CD19* L174V mutation and achieved prolonged remission of more than two years [53]. The ZUMA-2 trial demonstrated that among 60 patients with MCL, only one relapse was CD19-negative, while none of the patients who relapsed in ZUMA-5 were CD19-negative at disease progression [11]. In addition, in the ZUMA-1 trial, 75% of patients with loss of CD19 responded to Axicabtagene ciloleucel [53]. Thus, CD19-CAR T cell efficacy is also correlated with antigen-independent cytotoxicity [53], and the complexity of genomic and epigenomic alterations in lymphoma cells needs to be further analyzed to understand the alternative mechanisms of resistance to CD19-CAR T cell therapy.

#### 2.2.2. Epigenetic Downregulation

The tumoral epigenetic profile is an additional element that may confer resistance to novel immunotherapies, such as CD19-CAR T cells. IKAROS Family Zinc Finger 1 (*IKZF1*) is a gene that encodes a transcription factor belonging to the family of zinc-finger DNA-binding proteins associated with chromatin remodeling [54]. This protein acts as a regulator of lymphocyte differentiation, and overexpression of dominant-negative isoforms is linked to the onset of B-cell malignancies [54]. Domizi et al. demonstrated that knock-down of *IKZF1* in LCBL and CLL cell lines (respectively, OCI-Ly1, OCI-Ly7, SUDHL6; CI, JVM-2, WA-OSEL) is associated with reduced CD19 surface expression, as measured through flow cytometry, possibly leading to CD19-negative relapses [55]. This finding suggests that the epigenetic profile of B-cell lymphoma needs to be taken into consideration for preventing relapses, and additional studies are required to define further epigenetic alterations involved in the onset of resistance to CD19-CAR T cell therapy.

#### 2.2.3. Complex Genomic Alterations

*MYC* is a proto-oncogene that encodes a nuclear phosphoprotein involved in cell cycle progression, apoptosis, and cellular transformation, while *BCL2* encodes an integral outer mitochondrial membrane protein involved in the blockage of lymphocytes’ apoptosis process [56,57]. In addition, BCL6 is a zinc-finger transcription factor that modulates the transcription of STAT-dependent IL-4 responses of B cells and is involved in apoptosis and cell cycle control [58,59]. In a multicenter retrospective study conducted by Karmali et al., the survival outcomes of 408 adult patients with r/r DLBCL from 13 academic centers were evaluated after CD19-CAR T cell therapy [60]. DLBCL with increased expression of MYC (>40%) and BCL2 (>70%) analyzed through immunohistochemistry (IHC) and that without chromosomal rearrangements of *MYC* or *BCL2* analyzed through Fluorescence In Situ Hybridization (FISH) were defined as Double-Expressor Lymphomas (DELs), while Double-Hit Lymphomas (DHLs) were characterized by rearrangements of *MYC* and *BCL2* and/or *BCL6* irrespective of MYC or BCL2 expression [60]. The study showed that patients with DEL presented with the highest relapse rates after CD19-CAR T cell treatment, while DHL was linked to the worst OS following CD19-CAR T therapy [60]. Thus, *MYC*, *BCL2*, and *BCL6* overexpression and chromosomal rearrangements in aggressive B-cell lymphomas correlate with unfavorable prognosis after CD19-CAR T cell therapy. Despite these recent findings, the correlation between MYC-BCL2 double expression and CD19-CAR T cell resistance needs to be further elucidated, as this genomic alteration seems to represent a negative predictive index for newly diagnosed DLBCL rather than a mechanism of resistance specific to CAR T immunotherapy [61].

Furthermore, Jain et al. demonstrated through Whole-Genome Sequencing (WGS) of 51 lymphoma samples from 49 patients that complex genomic features may explain CD19-CAR T cell failure in DLBCL, transformed FL, and transformed CLL [53]. Apolipoprotein B mRNA-Editing Enzyme Catalytic Polypeptide-like (*APOBEC*) represents a family of cytosine deaminases that play a crucial role in innate immune defense, targeting viral DNA and RNA and contributing to lymphomagenesis through mutagenic activity detected in 7.8% of newly diagnosed DLBCL [53,62]. In relation to CD19-CAR T cell therapy response, single-base substitutions (SBSs) derived from APOBEC mutagenic activity were linked to significantly worse PFS in patients with LBCL, highlighting a strong connection between APOBEC activity and failure of treatment [53]. In addition, *RHOA* is a member of the RHO GTPases family involved in several cellular processes, such as cytoskeleton remodeling, cell adhesion, and motility [63]. WGS demonstrated that 10 out of 11 patients (91%) who progressed after CD19-CAR T cell therapy carried a focal-level deletion of *RHOA* (3p21.31) and that loss of *RHOA* leads to a diminished OS [53]. The molecular mechanism of resistance to CD19-CAR T cell therapy may be linked to increased motility of malignant and premalignant B lymphocytes, further leading to immune escape in tumoral niches where CAR T cell infiltration is limited [53]. Furthermore, Sworder et al. conducted a study on more than 700 tumor samples of r/r LBCL from two independent cohorts of patients (n_1_ = 65; n_2_ = 73) who received Axicabtagene ciloleucel [64]. The analysis revealed that alterations in genes involved in the definition of B cell identity (e.g., *PAX5*, *IRF8*), immune checkpoints (e.g., *PD-L1*), and the regulation of the TME (e.g., *TMEM30A*) are directly linked to CD19-CAR T cell resistance [64].

Lastly, chromothripsis is a severe form of genomic instability detected in around 20% of B-NHL, which consists of shattering and aberrant rearrangements of one or more chromosomes, leading to multiple aneuploidies [61,65]. In relation to this, 28 samples from patients with r/r DLBCL who received Axicabtagene ciloleucel and 50 newly diagnosed cases of DLBCL from the Pan-Cancer Analysis of Whole Genomes (PCAWG) were analyzed to understand the genomic drivers of resistance to CD19-CAR T cell therapy [61]. WGS revealed that 39.3% of r/r DLBCL was characterized by chromothripsis, which correlated with negative outcomes and early progression after treatment [61]. In addition, the *PPM1D* gene encodes for a member of the PP2C family of Ser/Thr protein phosphatases, and, in response to different environmental signals, its expression is induced in a p53-dependent way [66]. Molecular resistance to CD19-CAR T cell therapy in a cohort of 85 patients with r/r DLBCL was associated with mutations in *PPM1D* [67]. Gain-of-function mutations of PPM1D phosphatase confer independence from p53 activity with repercussion for impaired p53-dependent G1 checkpoints and cell proliferation [67]. Furthermore, the median PFS and OS of non-responders were significantly decreased (PFS of mutated vs. wild type patients: 3 vs. 12 months; OS: 5 vs. 37 months) [67]. These novel genomic findings reveal the complexity of molecular mechanisms underlying resistance to CD19-CAR T cell therapy, which reflect multifaceted interactions between lymphoma cells, T cells, and the TME [68].

#### 2.2.4. Lineage Switch

Lineage switching is an additional mechanism of resistance to CD19-CAR T cell therapy, usually described as the transition from Acute Lymphocytic Leukemia (ALL) to Acute Myeloid Leukemia (AML) or myeloid sarcoma upon relapse [8,69]. The conversion between lymphocytic and myeloid phenotype, associated with loss of gonadal lineage B antigens (such as CD19) and gain of myeloid molecular markers (e.g., cytoplasmic myeloperoxidase, CD64), represent an additional mechanism of immune escape, with two underlying causes described: immunoglobulin heavy chain reprogramming in myeloid stem cells and/or reprogramming/dedifferentiation of earlier B lymphoblastoid stem cells [8]. A few cases of lineage switching have been described in patients with B-cell lymphomas, including three cases of ML, one of DLBCL, and one of B-cell Lymphoblastic Lymphoma (B-LBL) [69]. After the initial diagnosis, these five B-cell lymphomas switched to AML with an M4 or M5 phenotype correlated with unfavorable outcomes [69]. Additional studies are needed to define the incidence of this phenomenon in B-cell lymphomas and its correlation with CD19-CAR T cell therapy; however, a previous analysis of B-ALL demonstrated that CD19-CAR T cell therapy may promote immune pressure leading to myeloid switching [70,71]. These data suggest that lineage switching may represent an alternative molecular mechanism of resistance to CD19-CAR T cell therapy in B-cell lymphomas.

### 2.3. Microenvironmental and Signaling Adaptations

Immune suppression mediated by the TME, metabolic competition, and lack of T cell infiltration decrease the efficacy of CD19-CAR T cell therapy [21,50,72]. Specifically, immunosuppressive cytokines, such as TGFβ, IL-10, and IL-6, and checkpoint regulators secreted in the TME are linked to decreased CAR T cell potency [32]. The definition of an immunosuppressive TME is complex, and various cellular subtypes are involved, such as osteoclasts, fibroblasts, endothelial cells, tumor-associated macrophages (TAMs), Myeloid-Derived Suppressor Cells (MDSCs), and regulatory T cells (Tregs) [73]. Jain et al. demonstrated that patients with LBCL who did not achieve durable remission after Axicabtagene ciloleucel had significant expression of interferon signaling, higher prevalence of MDSCs, and increased IL-6 and ferritin levels [74]. Firstly, interferon signaling is related to the expression of PD-L1 and MHC II, which are linked to early exhaustion by binding to PD-1 and LAG-3, and non-responder patients presented increased percentages of B cells positive for MHC II or PD-L1 [74]. Secondly, tumor interferon signaling inhibits CAR T cell proliferation, as demonstrated by Axicabtagene ciloleucel’s decreased expansion, and MDSCs contribute to this phenomenon [74]. Scholler et al. further analyzed the TME signature of 135 pretreatment and post-treatment tumor biopsies from 51 patients enrolled in the ZUMA-1 Phase 2 trial [75]. Within two weeks of Axicabtagene ciloleucel therapy, the expression of the cytotoxic T cell signature linked to CD8α and granzyme A genes, IFN-γ–regulated immune checkpoint encoding genes (*CD274*, *CD276*, *CTLA-4*), myeloid-related genes, and distinct chemokines (CD14, CD68, CCL2) was correlated with favorable responses in association with the elevation of IL-15, a well-known T cell growth factor [75]. In addition, CD8^+^ PD-1^+^ LAG-3^+/−^ TIM-3^−^ T cell density resulted in the strongest association with clinical efficacy [75]. Thus, the PD-1/PD-L1 pathway is firmly correlated with CAR T cell exhaustion, representing a crucial mechanism of immune escape [76]. Interestingly, Chen et al. analyzed tumor biopsies from the ZUMA-1 trial and found that CD19-CAR T was associated with less than 5% of T cells in the TME five days after Axicabtagene ciloleucel administration [77,78]. This finding suggests that CD19-CAR T cell therapy induces an immune response and recruits additional T populations strictly connected to favorable outcomes [77]. In conclusion, the TME is a key player in the onset of CD19-CAR T cell resistance, as it may be characterized by immunosuppressive signals, which directly influence various molecular pathways and cell interactions with an impact on the efficacy of this immunotherapy.

## 3. BsAbs and Resistance Onset

### 3.1. BsAbs’ Properties

Between 2022 and 2024, four BsAbs have been approved by the FDA and the EMA for the treatment of B-cell lymphomas, and their success is related to the recruitment of immune effector cells and the targeting of various molecular pathways [79]. BsAbs are dual-specificity molecules that bind two distinct epitopes simultaneously, and their mechanism of action is divided into combinatorial or obligate types [13,79]. Combinatorial BsAbs are composed of two independent extremities with dual inhibition of various targets, such as Receptor Tyrosine Kinases (RTKs) or Immune Checkpoint Inhibitors (CPIs) [79]. In contrast, the activity of obligate BsAbs depends on the simultaneous engagement or sequential temporal binding of the two extremities, as described for CD19 × CD3 BsAbs and transferrin receptor-based BsAbs [79]. BsAbs are produced through quadroma cell lines derived from two homologous hybridomas, chemical conjugation of two monoclonal antibody fragments, or genetic recombination [13]. Their role as therapeutics for hematologic malignancies is enhanced by the intrinsic interaction between tumor and immune cells, and, in this specific context, Bispecific T cell Engagers (BiTEs) represent the most promising BsAbs [13]. BiTEs are characterized by two binding sites for a precise antigen tumor (e.g., CD19, CD20, CD123, CD33, CD38) and CD3 on T cells. In 2022, Mosunetuzumab (CD20 × CD3) was the first approved BsAb for r/r FL, followed by the approval of Glofitamab (CD20 × CD3) and Epcoritamab (CD20 × CD3) for the treatment of r/r DLBCL [13,79]. BiTEs exert their action through the link between CD3^+^ CD4^+^/CD8^+^ T cells and Tumor-Associated Antigens (TAAs), with a cytotoxic effect mediated by the release of perforins and granzymes [80]. Their mechanism of action is further related to the recruitment of additional immune cells, and optimal T cell expansion is connected to peptide–MHC I stimulation, even though their cytotoxic effect is mainly MHC-independent [80]. Additional categories of BsAbs include Natural Killer (NK) Cell Engagers (BiKEs), BsAbs with immune checkpoint (ICP) modulation activity, and BsAbs, which block specific signaling pathways [80].

Various characteristics are linked to BsAbs’ functionality, including optimal target choice, epitope locations, grade of affinities, valencies, distance between binding sites, molecular size, ease of immunological synapse formation, balance between co-stimulatory and co-inhibitory signals, flexibility, and the presence of an Fc region associated with Fc-mediated effector functions [79,81]. Accordingly, two categories of BsAbs are now available, and this distinction influences their half-life and solubility [82]. The EMA- and FDA-approved BsAbs for B-cell lymphomas are IgG-like subtypes with an Fc region, characterized by higher molecular weights, prolonged half-life through the neonatal FcR (FcRn)-mediated recycling process, and the ability to induce Antibody-Dependent Cell-mediated Cytotoxicity (ADCC) compared to non-IgG-like subtypes with a molecular weight between 11 and 50 kDa [80,81,82]. However, limited permeability and tumor penetration, complex production compared to easy, large-scale production in microbial systems for the non-IgG-like subtype, and the onset of off-target interactions between Fc domains and Fcγ receptors represent the major disadvantages of this BsAb category [82]. Specifically, Mosunetuzumab exerts its action at low concentrations through the activation of CD69^+^ CD8^+^ T cells, with a maximum tumor depletion after 24 h of administration, while the decrease in its cytotoxic activity usually occurs after three days [81]. Glofitamab cytotoxic activity is demonstrated by the early formation of synapses between lymphoma and T cells, followed by efficient tumor lysis four hours after the encounter, whereas Epcoritamab determines dose-dependent activation of CD4^+^ and CD8^+^ T cells in association with perforin release [81]. Thus, the therapeutic index of BsAbs could be enhanced through the modification of various pharmacokinetic and pharmacodynamic parameters, including affinity, kinetics, and valency [82]. Table 2 summarizes the main characteristics of commercially available BsAbs for the treatment of B-cell lymphomas and highlights the major mechanisms of molecular resistance, which will be further discussed below.

### 3.2. T Cell Exhaustion and the Role of the TME

Primary and acquired resistance to BsAbs is connected to T cell dysfunction and early exhaustion due to the upregulation of immune checkpoint regulators [91]. The presence of M2 macrophages, immature MDSCs, and CD4^+^ CD25^+^ FOXP3^+^ Tregs switches the TME to an immunosuppressive milieu, which further promotes T cells’ anergy and apoptosis [92]. Additionally, immunosuppressive cytokines, such as IL-10 and TGFβ, and specific enzymes, like indoleamine-2,3-dioxygenase 1 (IDO1), induce weaker T cell infiltration and lead to secondary T cells’ terminal differentiation and failure [82,92]. Furthermore, early T cell exhaustion is induced by tonic TCR activation and CD3 signaling, which are connected to prolonged and continuous exposure to BsAbs; thus, optimal administration should be considered to avoid this phenomenon [82]. Following continuous TCR triggering, CD3 downregulation is a common mechanism of resistance to CD20 × CD3 BsAbs, and prolonged T cell activation leads to the upregulation of inhibitory molecules, such as PD-1, TIM3, and LAG3, affecting T cells’ proliferation [81,82]. Lastly, the recruitment of polyclonal T cell populations may be counterproductive, as demonstrated by the negative effect of CD3^+^ CD4^+^ CD25^high^ FOXP3^+^ Tregs in patients with precursor B-ALL [93]. Following Blinatumomab (CD19 × CD3 BsAbs) administration, T cells’ proliferation was negatively affected by this cellular population, and CD8-mediated cytotoxicity was diminished, suggesting that more selective recruitment of T cells could contribute to decreased onset of resistance to BsAbs [93]. T cell fitness is additionally impaired by previous therapeutic regimens; thus, the efficacy of BsAbs in heavily pretreated patients decreases [92]. These findings demonstrate that the TME plays a crucial role in the definition of T cell cytotoxicity; however, additional studies on B-cell lymphomas and CD20 × CD3 BsAbs are needed to clarify the efficacy and fitness of recruited T cells to provide successful therapeutic interventions.

### 3.3. Tumor-Intrinsic Factors

#### 3.3.1. CD20 Antigen Loss and Alterations

In relation to r/r B-NHL, anti-CD20 therapies are used in various lines of treatment and represent optimal therapeutic regimens for CD20-positive malignancies. The CD20 gene (*MS4A1*) is localized in 11q12 and encodes a B-lymphocyte surface molecule, specifically a member of the membrane-spanning 4A gene family, which is involved in the development and differentiation of B cells into plasma cells [94]. Rituximab is a humanized monoclonal antibody that selectively targets CD20 and leads to Complement-Mediated Cytotoxicity (CMC) and ADCC, with additional indirect effects spanning sensitization of cancer cells to chemotherapy and apoptosis induction [95]. Specifically, in third-line settings, elderly or frail patients with r/r FL who achieve long-term remission are suitable candidates for Rituximab monotherapy as a rescue approach [5]. Furthermore, Obinutuzumab is an additional monoclonal antibody that exerts its mechanism of action by binding CD20 on lymphoma cells and inducing ADCC [96]. In combination with Lenalidomide, it has been shown that Obinutuzumab induced complete remission in a patient with r/r FL that progressed after two CAR T cell administrations [97]. These findings demonstrate that anti-CD20 immunotherapies represent a solid therapeutic option not only in the first lines of treatment but also after CAR T cell therapy failure.

In this specific context, as described for CD19-CAR T cell therapy, a frequent mechanism of immune escape is represented by antigen loss. Grigg et al. analyzed 41 patients with r/r DLBCL and PMBCL following Glofitamab treatment after a median of three previous lines of therapy and found that at the moment of relapse, the rate of CD20 loss at progression was 59%, with a median OS of 4.1 months after relapse [80,98]. In addition, Schuster et al. performed IHC, RNA sequencing, and Whole-Exome Sequencing (WES) on tumor samples from patients with r/r B-NHL treated with Mosunetuzumab monotherapy to evaluate CD20 expression [99]. Prior to treatment, 10.9% of patients had intermediate levels of CD20 (10–74%), whereas a small subset (5.5%) was characterized by inferior levels (<10%), which correlated with unfavorable prognosis, suggesting that the evaluation of CD20 expression should be considered before treatment [99]. WES revealed 14 CD20 variants derived from distinct mutations in the extracellular loop and the transmembrane domain, but the latter were not associated with decreased CD20 expression [99]. However, two missense mutations (C167G, K175E) in the extracellular loop 2 (ECL2), which contains the therapeutic binding site for anti-CD20 therapy, correlated with an absence of response, even though CD20 expression was preserved [99]. Rushton et al. further analyzed missense mutations affecting *MS4A1* and registered a correlation between rapid DLBCL recurrence and reduced CD20 expression and stability following Rituximab administration, suggesting that previous anti-CD20 therapeutic regimens could affect CD20 × CD3 BsAbs’ efficacy [100]. After identifying four variants (V1-V4) with distinct 5′ untranslated regions (UTR), Ang et al. further discovered that CD20 splicing determines resistance to Mosunetuzumab [101]. The shift between V3 and V1 determined the downregulation of CD20 in relapsed FL, indicating that alternative splicing plays a crucial role in some CD20-negative relapses [101]. The latter are also linked to the deletion of *MS4A1*, which correlates with unfavorable prognosis [102,103]. With a median follow-up of 18.3 months and following anti-CD20 therapy, Michot et al. demonstrated that CD20 negativity was linked to a decreased OS of 8.9 months (2.4–19.1, 95% CI) compared to CD20-positive r/r FL, with an OS of 28.3 months (25.1–75.3, 95% IC) [103]. Additionally, Duell et al. demonstrated through WES that truncating mutations of *MS4A1* affected 80% of patients who progressed after CD20 × CD3 BsAbs therapy [104]. Despite these recent findings, further analysis of CD20 expression is needed, as clinical responses to approved BsAbs are observed in patients with progressive disease [105]. Accordingly, Schuster et al. analyzed a cohort of 293 patients with r/r B-NHL who received Mosunetuzumab in three-week cycles and found out that CD20 expression was preserved in the majority of patients with progressive disease, suggesting that alternative mechanisms of resistance should explain these recurrences, as described for CD19-positive relapses following CAR T cell therapy [106].

#### 3.3.2. Tumor Genomic Alterations

Novel molecular markers have been identified as responsible for BsAbs’ resistance onset in B-cell lymphomas. Firstly, resistance to Glofitamab is linked to the downregulation of p53 targets and the overexpression of MYC ones, while increased frequency of *TP53* mutations was identified in patients with DLBCL progression [81,92]. Additionally, Kyvsgaard et al. were able to detect pretreatment genomic alterations in 56 patients with B-NHL who received CD20 × CD3 BsAbs between 2017 and 2023 by performing Next Generation Sequencing (NGS) with a custom lymphoma panel [15]. Genetic alterations of *NOTCH1* were linked to limited survival, and increased proliferation of aberrant clones was observed during CD20 × CD3 BsAb therapy, suggesting a possible role in the onset of BsAbs’ resistance [15]. Lastly, Zucchinetti et al. analyzed a cohort of 41 patients with r/r LBCL treated with Glofitamab and performed CAPP sequencing of circulating tumor DNA (ctDNA), demonstrating that more than 21% of r/r LBCL presented with *TP53* (44%), *KMT2D*, *PIM1* and *IGLL5* (37% each), *CARD11* (27%), *HIST1H1E* and *CREBBP* (24% each), or *BCL2* (22%) mutations [107]. *TP53* mutations (44%, 18/41 patients) were not predictive of worse PFS; however, their persistence after the third cycle of therapy was indicative of progression [107]. In order to provide efficient therapeutic options, further genetic and epigenetic analysis are needed to identify additional drivers of resistance to BsAbs in B-cell lymphomas.

## 4. Conclusions

EMA- and FDA-approved CD19-CAR T cells and BsAbs for the treatment of B-cell lymphomas are improving patients’ survival after failure of previous standard-of-care treatment; however, the onset of molecular resistance represents a threat to their clinical efficacy. T cell characteristics prior to CD19-CAR T cell manufacturing, such as their phenotype and epigenetic profile, influence proliferation rates and are linked to early exhaustion. T cell fitness is equally essential to the success of CD20 × CD3 BsAbs, which relies on the recruitment of wide populations of T cells. Persistent TCR signaling, CD3 downregulation, and PD-1, TIM3, and LAG3 upregulation contribute to early T cell exhaustion and decreased proliferation. An alternative mechanism of resistance to CD19-CAR T cells and BsAbs is antigen modulation. CD19 and CD20 are frequently downregulated, and alternative splicing, in association with genetic mutations, is a common mechanism of immune escape. In addition, complex genomic alterations play a crucial role in the onset of molecular resistance to CD19-CAR T cell and BsAb therapy. Lastly, the immunosuppressive TME may affect distinct molecular pathways and contribute to early exhaustion of CAR T and T cells. Immune checkpoint inhibitors, such as anti-PD-1 (e.g., Pembrolizumab) and anti-PD-L1 (e.g., Atezolizumab) treatments, improve immunotherapy’s efficacy in combination with CD19-CAR T cell and CD20 × CD3 BsAbs. Thus, they represent major therapeutic strategies for overcoming immune escape. Furthermore, personalized targeted therapies focused on specific genomic alterations that confer resistance to CD19-CAR T cell and BsAb therapies should be considered, and the development of novel therapeutic approaches should pave the way towards a new era of cancer immunotherapy. These findings illustrate the complexity of molecular mechanisms underlying failure of the described therapeutics and demonstrate the necessity of innovative analysis, which could reveal additional drivers of resistance to provide successful therapies in the clinical setting.

## Figures and Tables

**Table 1 cancers-17-03453-t001:** EMA- and FDA-approved CD19-CAR T cell therapies for B-cell lymphomas.

Trade Name (Product Name)	Target	Construct	Approval	Indications *	Molecular Mechanisms of Resistance
Kymriah [40,41,42] (Tisagenlecleucel)	CD19	CD8α/CD8α 4-1BB + CD3ζ	2017 (FDA) 2018 (EMA)	Adult patients with r/r DLBCL, DLBCL arising from FL, FL, or High-Grade B-Cell Lymphoma (HGBCL) after two or more lines of therapy	T cell determinants ● Phenotype, polyfunctionality ● CAR construct ● Epigenetic modifications (e.g., *TET2*, *DNMT3A*) Tumor-intrinsic factors ● CD19 reduced density and mutations (e.g., exon 2 and 3) ● Alternative splicing, epigenetic modifications (e.g., *IKZFN1*) ● Gene overexpression (e.g., *MYC*, *BCL2*), chromosomal rearrangements (e.g., *BCL6*) ● *PAX5*, *IRF8*, *PD-L1*, *TMEM30A,* and *PPM1D* mutations, *RHOA* deletion, APOBEC mutagenic activity ● Trogocytosis, hyperglycosilation, antigen shedding, chromothripsis, lineage switch TME features ● Immunosuppressive cells (e.g., TAMs, MDSCs, Tregs) ● Immunosuppressive cytokines (e.g., TGFβ, IL-10, IL-6) ● Inhibitory pathways (e.g., PD-1/PD-L1)
Yescarta [40,43,44] (Axicabtagene ciloleucel)	CD19	CD28/CD28 CD28 + CD3ζ	2017 (FDA) 2018 (EMA)	Adult patients with r/r DLBCL, FL, HGBCL, or Primary Mediastinal B-Cell Lymphoma (PMBCL) after first-line therapy	
Tecartus [40,45,46] (Brexucabtagene autoleucel)	CD19	CD28/CD28 CD28 + CD3ζ	2020 (EMA, FDA)	Adult patients with r/r MCL after two or more lines of therapy	
Breyanzi [40,47,48] (Lisocabtagene maraleucel)	CD19	IgG4/CD28 4-1BB + CD3ζ	2021 (FDA) 2022 (EMA)	Adult patients with r/r HGBCL, FL grade 3B, MCL, PMBCL, CLL, or Small Lymphocytic Lymphoma (SLL) after first-line therapy	

* Disease stage and patients’ age and health status influence the choice of therapy. First lines of treatment for DLBCL include R-CHOP (Rituximab, Cyclophosphamide, Doxorubicin, Vincristine, Prednisone) and its variants, ISRT (Involved Site Radiation Therapy) 30 Gy, and anthracyclines, while second lines of therapy are represented by platinum-based regimens, Autologous Stem Cell Transplantation (ASCT), and a combination of novel antibodies with conventional chemotherapy. First lines of treatment for FL include ISRT 24 Gy, R-CHOP, and Rituximab monotherapy or in combination with other drugs, such as Bendamustine and Lenalidomide. Second lines of treatment include chemotherapy, anti-CD20 maintenance, and Rituximab–Lenalidomide with or without Tafasitamab. MCL treatment combines ISRT, covalent Bruton Tyrosine Kinase inhibitors (cBTKi), and chemotherapy regimens, including Rituximab and Dexamethasone/Prednisone. Before administering CAR T cell therapy, cBTKis with or without Venetoclax should be considered [5]. First lines of therapy for CLL include BTKis, Venetoclax in association with Obinutuzumab, and anti-CD20-based chemoimmunotherapy, while second lines of treatment include BTKis, Venetoclax and Rituximab, and Idelalisib associated with Rituximab and Allogeneic SCT [49]. HGBCL and PMBCL are subtypes of DLBCL and are therefore subjected to similar treatment guidelines, while SLL guidelines follow CLL protocols.

**Table 2 cancers-17-03453-t002:** EMA- and FDA-approved bispecific antibodies for B-cell lymphomas.

Trade Name (Product Name)	Target (CD20:CD3 Ratio)	Structure	Approval	Indications	Molecular Mechanisms of Resistance
Lunsumio [83,84,85] (Mosunetuzumab)	CD20 × CD3 (1:1)	IgG1	2022 (EMA, FDA)	Adult patients with r/r FL after two or more lines of therapy	BsAbs properties ● Pharmacokinetic and pharmacodynamic characteristics TME features ● Immunosuppressive cells (e.g., MDSCs, Tregs, M2 macrophages) ● Immunosuppressive cytokines (e.g., IL-10, TGFβ) T cell dysfunction ● TCR tonic signaling ● CD3 downregulation ● Previous therapeutic regimens Tumor-intrinsic factors ● *MS4A1* mutations or downregulation ● Alternative splicing ● *TP53* downregulation and mutations ● MYC target genes’ overexpression ● *NOTCH1*, *KMT2D*, *PIM1*, *IGLL5*, *CARD11*, *HIST1H1E*, *CREBBP*, and *BCL2* mutations
EMA: Tepkinly FDA: Epkinly [83,86,87] (Epcoritamab)	CD20 × CD3 (1:1)	IgG1	2023 (EMA, FDA)	Adult patients with r/r DLBCL, FL, or HGBCL after two or more lines of therapy	
Columvi [83,88,89] (Glofitamab)	CD20 × CD3 (2:1)	IgG1	2023 (EMA, FDA)	Adult patients with r/r DLBCL or r/r HGBCL from FL after two or more lines of therapy	
Ordspono [83,90] (Odronextamab)	CD20 × CD3 (1:1)	IgG4	2024 (EMA)	Adult patients with r/r DLBCL or FL after two or more lines of therapy	

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
