# Peer review of "Mechanisms of Resistance to Novel Immunotherapies in B-Cell Lymphomas: Focus on CAR T and Bispecific Antibodies"

_cancers, 2025, doi:10.3390/cancers17213453_

Round 1

Reviewer 1 Report

Comments and Suggestions for Authors

The review article by Arena et al focuses on the mechanisms of resistance to novel immunotherapies (CAR-T and bispecific antibodies) in B-cell lymphomas.

Major comments

  1. It would be beneficial if the conclusion of the article includes discussion about the importance of personalized targeted therapies since Cancer cells can utilize mechanisms discussed to evade immunosurveillance.

  1. To provide your readers with better context, please consider in section 3.3.1 to include conventional anti-CD20 therapies used after CAR-T failure, such as rituximab and obinutuzumab have been explored, often in combination with other therapies before proceeding to the rest of the discussion with Glofitamab etc.

  1. Please consider adding a column to table 1 and 2 to bullet point the mechanisms of resistance. This allows the reader to quickly understand key points in the review.

  1. Please consider adding a paragraph pertaining to the role of conditioning regimen administered before CAR T-cell therapy in CAR T-cell resistance. The specific regimen with their doses timing impacts the effectiveness of the therapy and can influence whether a patient's cancer develops resistance

Minor comments

  1. Cumbersome abbreviations: Pages of abbreviations may not be necessary since your target readers already be very well versed in the basic science field.

  1. Issue with writing style: Please consider introducing the meaning scientific terms before and not after an idea was introduced. For example, (lines 281-283) “…that loss of RHOA leads to a diminished OD [57]. RHOA is a member of the RHO GTPases family in several cellular processes.” Another example, (lines 295-297) “ r/r DLBCL was characterized by chromothripsis, which …[65]. Chromothripsis is a sever form of genome instability..

Author Response

Comment 1: "It would be beneficial if the conclusion of the article includes discussion about the importance of personalized targeted therapies since Cancer cells can utilize mechanisms discussed to evade immunosurveillance."

Response 1: We agree with this suggestion and therefore added an explanation of personalized targeted therapies which represent major strategies to overcome immune escape (lines 564-570).

Comment 2: "To provide your readers with better context, please consider in section 3.3.1 to include conventional anti-CD20 therapies used after CAR-T failure, such as rituximab and obinutuzumab have been explored, often in combination with other therapies before proceeding to the rest of the discussion with Glofitamab etc."

Response 2: That's a good point, therefore we added a brief description of anti-CD20 therapies following CAR-T failure to accomplish this goal (lines 478-493).

Comment 3: "Please consider adding a column to table 1 and 2 to bullet point the mechanisms of resistance. This allows the reader to quickly understand key points in the review."

Response 3: Following this suggestion, we modified Table 1 and 2 by adding the relative mechanisms of resistance.

Comment 4: "Please consider adding a paragraph pertaining to the role of conditioning regimen administered before CAR T-cell therapy in CAR T-cell resistance. The specific regimen with their doses timing impacts the effectiveness of the therapy and can influence whether a patient's cancer develops resistance"

Response 4: Between lines 121 and 125, we explained that the quality of autologous T cell is a crucial element which influences CD19-CAR T cell efficiency and is linked to previous therapeutic regimens. Following this comment, we provided a more direct explanation of how conditioning regimens affect CAR T cell effectiveness (lines 126-148).

Comment 5: "Cumbersome abbreviations: Pages of abbreviations may not be necessary since your target readers already be very well versed in the basic science field."

Response 5: Thank you for your minor comment. We decided to include a broad list of abbreviations used in the review according to the journal policy and previous published reviews on Cancers.

Comment 6: "Issue with writing style: Please consider introducing the meaning scientific terms before and not after an idea was introduced. For example, (lines 281-283) “…that loss of RHOA leads to a diminished OD [57]. RHOA is a member of the RHO GTPases family in several cellular processes.” Another example, (lines 295-297) “ r/r DLBCL was characterized by chromothripsis, which …[65]. Chromothripsis is a sever form of genome instability.."

Response 6: This is a great suggestion and therefore we moved the scientific descriptions before explaining our findings throughout the manuscript.

Thank you for your useful comments and kind suggestions. Please see the attached document to check the amendments highlighted in yellow.

Reviewer 2 Report

Comments and Suggestions for Authors

In this review, Arena and Chiarle summarize the current molecular mechanisms of resistance to CAR-T and CD3×tumor bispecific antibody (BsAb) therapies. They provide an in-depth review of how host immune features, tumor-intrinsic determinants, and the tumor microenvironment shape resistance to therapies. The manuscript is well organized, conceptually coherent, and timely, and I’m confident it will benefit a broad readership. I only have a few minor suggestions to improve clarity and consistency.

  1. Please clarify what is meant by first- and second-line therapy in Table 1 by including a brief footnote.
  2. Add explicit in-text references to Tables 1 and 2 at their first mention. Also, please move Table 2 into Section 3 where BsAbs are discussed.
  3. Can authors further clarify the mechanism of action for BsAbs after the dual engagement of CD3 and B cell antigen on lymphoma cells?
  4. The text states that nine BsAbs are approved in the US/EU (ref. 83), but Table 2 lists four, and ref. 28 is a review. If Table 2 is intended to show representative BsAbs, please state this explicitly in the text; for each BsAb listed in Table 2, can authors match it to a primary approval source by FDA/ EMA?
  5. For each CD19 CAR-T therapy listed in Table 1, can authors match it to a primary approval source by FDA/ EMA?
  6. Table 2, replace CD xx with the actual antigen name.
  7. Please italicize gene symbols throughout the manuscript. For example, MYC and BCL2 in lines 255-256; PPM1D at line 301.
  8. Line 51: “100,000 per year and 85,000 new diagnosis”; line 194, “more than 1,000 antigens per tumor cell”.

Author Response

Comment 1: "Please clarify what is meant by first- and second-line therapy in Table 1 by including a brief footnote."

Response 1: We agree with this comment and therefore added the relative footnote (lines 205-216).

Comment 2: "Add explicit in-text references to Tables 1 and 2 at their first mention. Also, please move Table 2 into Section 3 where BsAbs are discussed."

Response 2: Thank you for this suggestion. We proceeded to add in-text references to Table 1 and 2 (line 201 and 447) and we moved the latter to BsAbs section (line 450).

Comment 3: "Can authors further clarify the mechanism of action for BsAbs after the dual engagement of CD3 and B cell antigen on lymphoma cells?"

Response 3: Following this comment, we further explained the mechanisms of action for BsAbs between lines 417 and 424.

Comment 4: "The text states that nine BsAbs are approved in the US/EU (ref. 83), but Table 2 lists four, and ref. 28 is a review. If Table 2 is intended to show representative BsAbs, please state this explicitly in the text; for each BsAb listed in Table 2, can authors match it to a primary approval source by FDA/ EMA?"

Response 4: We agreed to clarify this concept and thus wrote a clear explanation of it (lines 399-401). Table 2 is intended to describe the commercially approved BsAbs (line 450) and we added a clarification for this in the main text (lines 447-449). We further matched every therapeutics with its own requested source.

Comment 5: "For each CD19 CAR-T therapy listed in Table 1, can authors match it to a primary approval source by FDA/ EMA?"

Response 5: We agreed on this comment and matched every single CAR T product with its primary approval source.

Comment 6: "Table 2, replace CD xx with the actual antigen name."

Response 6: We proceeded to replace CD** with CD20 in Table 2.

Comment 7: "Please italicize gene symbols throughout the manuscript. For example, MYC and BCL2 in lines 255-256; PPM1D at line 301."

Response 7: Following this annotation, we modified every gene symbols in the manuscript as indicated.

Comment 8: "Line 51: “100,000 per year and 85,000 new diagnosis”; line 194, “more than 1,000 antigens per tumor cell""

Response 8: Thank you for pointing this out. We changed the sentences according to your suggestion.

Thank you again for your kind suggestions and useful comments. Please, see the attached document to check the modifications highlighted in green.

Reviewer 3 Report

Comments and Suggestions for Authors

This is a very well written, comprehensive and timely review that discusses mechanisms of resistance to immunotherapy with CAR T cells and bispecific Abs of B cell lymphomas. The authors call attention to the fact that resistance to  immunotherapy (IT) of B cell lymphomas may be seen in as many as 40-50% of cases. They emphasize that urgent need exists for (a) better understanding of molecular/genetic mechanisms responsible for resistance and (b) better definition of factors /mechanisms that may mitigate resistance to IT. The review consists of two parts: one devoted to immune evasion that arises in response to CD19-CART cell therapy and the other is focused on resistance to bispecific Abs. Throughout the text, the authors systematically and thoroughly and thoughtfully discuss various diverse mechanisms of resistance to IT.     In conclusion, they stress that T cell fitness prior to and throughout therapy may the key to success. The latter is clearly linked the TME and its immunosuppressive effects on immune cells. Unfortunately, we remain largely unsuccessful in various attempts to ameliorate immune suppression in the TME and simultaneously optimize T cell fitness. The review does not provide an answer, but it realistically discusses potential barriers       and opportunities that exist or may emerge in the near future. In this respect, the review is an important contribution to the field of IT in cancer, including lymphoid malignancies. 

Author Response

Comment 1: "This is a very well written, comprehensive and timely review that discusses mechanisms of resistance to immunotherapy with CAR T cells and bispecific Abs of B cell lymphomas. The authors call attention to the fact that resistance to  immunotherapy (IT) of B cell lymphomas may be seen in as many as 40-50% of cases. They emphasize that urgent need exists for (a) better understanding of molecular/genetic mechanisms responsible for resistance and (b) better definition of factors /mechanisms that may mitigate resistance to IT. The review consists of two parts: one devoted to immune evasion that arises in response to CD19-CART cell therapy and the other is focused on resistance to bispecific Abs. Throughout the text, the authors systematically and thoroughly and thoughtfully discuss various diverse mechanisms of resistance to IT.     In conclusion, they stress that T cell fitness prior to and throughout therapy may the key to success. The latter is clearly linked the TME and its immunosuppressive effects on immune cells. Unfortunately, we remain largely unsuccessful in various attempts to ameliorate immune suppression in the TME and simultaneously optimize T cell fitness. The review does not provide an answer, but it realistically discusses potential barriers       and opportunities that exist or may emerge in the near future. In this respect, the review is an important contribution to the field of IT in cancer, including lymphoid malignancies. "

Response 1: Thank you for your useful comment and kind reply.